# Perceived Barriers and Intentions to Receive COVID-19 Vaccines: Psychological Distress as a Moderator

**DOI:** 10.3390/vaccines11020289

**Published:** 2023-01-28

**Authors:** Ngo Thi Viet Nga, Vu Ngoc Xuan, Vu Anh Trong, Pham Huong Thao, Duong Cong Doanh

**Affiliations:** National Economics University, Hanoi 100000, Vietnam

**Keywords:** psychological distress, perceived barriers, self-efficacy, attitude towards COVID-19 vaccines, intention to receive COVID-19 vaccines

## Abstract

A high vaccination uptake degree is crucial to achieve herd immunity to COVID-19 and restrict the spread of the COVID-19 pandemic. However, little is known about the antecedents that reduce or contribute to shaping the intention to receive COVID-19 vaccines, as well as how psychological distress—a mental health problem—can reinforce or dampen the translation from antecedents into intention to receive COVID-19 vaccines. The objective of this study was to explore the effects of perceived clinical and access barriers, self-efficacy, and attitudes towards COVID-19 vaccines on the intention to receive COVID-19 vaccines. Simultaneously, the moderation effects of psychological distress on this relationship were also examined. Using a sample of 2722 Vietnamese adults and structural equation modeling (SEM), this study illustrated that self-efficacy and attitudes towards COVID-19 vaccines were significantly interrelated with intention to receive COVID-19 vaccines. Self-efficacy, attitudes towards COVID-19 vaccines, and intention to receive COVID-19 vaccines were negatively influenced by perceived access barriers but were positively associated with perceived clinical barriers. Importantly, our study reported that when psychological distress was higher, the link between self-efficacy and intention to receive COVID-19 vaccines will become weaker, but the effect of perceived clinical barriers on intention to receive COVID-19 vaccines will be reinforced. Moreover, self-efficacy and attitudes towards COVID-19 vaccines served as mediators in the linkages between perceived barriers and intention to receive COVID-19 vaccines. Besides providing contributions to the extant COVID-19 vaccine literature, this study provides useful recommendations for practitioners and policymakers to foster adults’ COVID-19 vaccine acceptance.

## 1. Introduction

The novel coronavirus disease 2019 (COVID-19), which originated in Wuhan City, China, has resulted in catastrophic damages worldwide [1,2,3]. More than 200 countries around the world have reported infection cases and deaths. As of 15 January 2023, there were more than 661 million total confirmed cases, with more than 6.7 million total deaths [4,5]. The COVID-19 pandemic has caused massive losses and socioeconomic panic worldwide [6,7,8]. In Vietnam, 11,525,711 infected people and 43,186 deaths were reported while the number of administered vaccine doses accounted for 265,518,865 [9]. However, to date, an effective treatment of COVID-19 disease is still not available; thus, almost all countries are dependent on preventative methods to limit the spread of the COVID-19 pandemic [10,11]. Vaccines have been determined as one of the most efficient preventive measures to control the spread of infectious diseases [12,13,14]. The current extensive efforts devoted to vaccine research and development in response to the COVID-19 pandemic are considered unprecedented regarding speed and scale [15]. Thus far, there have been a number of COVID-19 vaccines approved for use by WHO, including Moderna-USA, Prizer/BioNTech-USA, Janssen-USA, Oxford/AstraZeneca- United Kingdom, Covishield- USA, Bharat Biotech, Sinopharm and Sinovac-Beijing, China. Moreover, hundreds of vaccine candidates and ongoing vaccine trials are being pursued [16]. However, the phenomenon of vaccine hesitancy has been widely documented and the problem of vaccine refusal is present worldwide, despite available vaccines [12,13]. A high vaccination uptake level is necessary to achieve herd immunity during the COVID-19 pandemic [13]. A high level of COVID-19 vaccine refusal would result in failure to achieve herd immunity and thus the containment of the COVID-19 pandemic. Therefore, understanding adults’ intention towards COVID-19 vaccination is crucial for the successful implementation of a COVID-19 vaccination campaign [13]. There have been numerous studies that have investigated the issue of COVID-19 vaccine intention and hesitancy in many countries, such as the USA [3], the United Kingdom, Canada [17], India [18] and others [19]. Noticeably, although some studies have been conducted to explore the issues of willingness to get vaccinated against COVID-19 and/or COVID-19 vaccine acceptance in Southeast Asian countries, including Vietnam [7,8,10], attention paid to explore the antecedents of vaccination intention and hesitancy is still scant. 

Moreover, although several prior studies have investigated the impacts of negative and positive antecedents on intention/hesitancy to receive COVID-19 vaccines [11,13,14], our understanding of the moderation effects of mental health problems, such as psychological distress, on COVID-19 vaccine acceptance and uptake is still limited [19]. Thus, this research aimed to bridge the literature gap by testing the moderation impacts of psychological distress on the paths from perceived clinical and access barriers, self-efficacy, and attitude towards COVID-19 vaccines to intention to receive COVID-19 vaccines. Simultaneously, the mediating role of self-efficacy and attitudes towards COVID-19 vaccines in the links between perceived barriers and intention to receive COVID-19 vaccines was also tested in our study.

## 2. Conceptual Framework and Hypotheses 

### 2.1. Attitudes towards COVID-19 Vaccines

Attitudes refer to how a person consider something and tends to act towards it, often in an evaluative way [13,20]. Theory of planned behavior [21] argue that attitude towards behavior is the best predictor of behavioral intentions. Indeed, a number of prior studies, which applied the theory of planned behavior in their hypothesized frameworks, also reported that attitudes towards behaviors is determined as the most influential antecedents of behavioral intentions [21,22,23]. In the research area of COVID-19 vaccine acceptance, some previous studies also revealed that individuals’ attitudes towards COVID-19 vaccines was found to be positively related to their intentions to receive COVID-19 vaccines [20]. In the context of Vietnam, it is hypothetical that attitudes towards COVID-19 vaccines play the crucial role in predicting intentions to receive COVID-19 vaccines among Vietnamese adults. 

**H1:** 
*Attitudes towards COVID-19 vaccines is significantly related to intentions to receive COVID-19 vaccines.*


### 2.2. Self-Efficacy

Acknowledging that actions related to health issues, such as receiving COVID-19 vaccines to prevent the unexpected consequences might not be sufficient to motivate individuals carry out these behaviors [11]. Self-efficacy, which reflects the perceived capacity of being vaccinated with COVID-19 vaccines, can play the crucial important role in executing such behaviors. Moreover, some scholars also highlight that self-efficacy can be seen as a similar construct in theory of planned behavior [21,24], namely perceived behavior control, while perceived behavior control was found to be a strong predictor of attitude towards behavior and behavioral control [11,23,25]. Consequently, self-efficacy can be significantly correlated with attitudes towards COVID-19 vaccines and intentions to COVID-19 vaccines among Vietnamese adults. The following hypotheses are therefore formulated. 

**H2:** 
*Self-efficacy is significantly related to (a) attitudes towards COVID-19 vaccines and (b) intentions to receive COVID-19 vaccines.*


### 2.3. Perceived Barriers 

Coe et al. [26] defined perceived barriers as the beliefs related to the efficacy and the costs of the expected actions. They can be divided into perceived clinical barriers and perceived access barriers to vaccination [26]. Empirically, some studies investigated how the perceptions of different barriers to intentions or/and willingness to receive COVID-19 vaccines [12,26]. However, no prior studies considered perceived barriers as two separated constructs and their effects on intentions to receive COVID-19 vaccines. In this study, we suppose that perceived clinical and access barriers can affect significantly self-efficacy, attitudes towards COVID-19 vaccines, and intentions to receive COVID-19 vaccines. In other words, when people perceive barriers related to the safety, side effects, as well as the difficulty level of being vaccinated with COVID-19 vaccines, their intentions to receive COVID-19 vaccines can be influenced. Moreover, Chu and Liu [11] argue that self-efficacy and attitudes towards COVID-19 vaccines can receive the effects of different factors, such as cues to actions, descriptive and injunctive norms, then transfer these impacts on intentions to receive COVID-19 vaccines. Thus, besides direct effects, individuals perceived clinical and access barriers can also indirectly influence their intentions to receive COVID-19 vaccines through self-efficacy and attitudes towards COVID-19 vaccines. Consequently, in the context of Vietnam, some following hypotheses are proposed. 

**H3:** 
*Perceived clinical barriers are significantly correlated with (a) self-efficacy, (b) attitudes towards COVID-19 vaccines, and (c) intentions to receive COVID-19 vaccines.*


**H4:** 
*Perceived access barriers are significantly correlated with (a) self-efficacy, (b) attitudes towards COVID-19 vaccines, and (c) intentions to receive COVID-19 vaccines.*


### 2.4. Psychological Distress 

Psychological distress reflects the unpleasant emotional reactions to stress states, which are unmanaged and overwhelmed [27]. Almost all prior studies only considered psychological distress as the detrimental factor of life satisfaction and/or well-being [28,29], while neglecting the moderation effect of psychological distress. In this study, we suppose that psychological distress can serve as the moderator which may facilitate or weaken the impacts of self-efficacy, attitudes towards COVID-19 vaccines, and perceived clinical and access barriers on intentions to receive COVID-19 vaccines. Particularly, the transformations from self-efficacy, attitudes towards COVID-19 vaccines, and perceived clinical and access barriers into intentions to receive COVID-19 vaccines can be strengthened or weakened depending on the high or low degree of psychological distress. In the context of Vietnam, it is hypothetical that psychological distress can significantly moderate the relationship between self-efficacy, attitudes towards COVID-19 vaccines, perceived clinical and access barriers, and intentions to receive COVID-19 vaccines among Vietnamese adults.

**H5:** 
*Psychological distress significantly moderates the effects of (a) attitude towards COVID-19 vaccines and (b) self-efficacy on intentions to receive COVID-19 vaccines.*


**H6:** 
*Psychological distress significantly moderates the effects of (a) perceived clinical barriers and (b) perceived access barriers on intentions to receive COVID-19 vaccines.*


Figure 1 illustrates the conceptual framework of this study. 

## 3. Materials and Methods 

### 3.1. Scale and Questionnaire Development 

To test the moderation impacts of psychological distress on the association between perceived clinical and access barriers, self-efficacy, attitudes towards COVID-19 vaccines, and intention to receive COVID-19 vaccines, a questionnaire survey was utilized in our study to collect the dataset. The survey and questionnaire is listed in the Appendix A. The Table 1 is presented Demographic characteristics of respondents. All scales used in our study were modified from prior studies. Specifically, the first three items (ICV1, ICV2, and ICV3) of the scale regarding “intention to receive COVID-19 vaccines” were adopted from Chu and Liu [11,30,31], while the last items (ICV4, ICV5, ICV6, ICV7, ICV8) of this scale were modified from Mir et al. [13,32,33]. The five-item scale to assess “attitudes towards receiving COVID-19 vaccines” was modified from Mir et al. [13,34,35,36]. The three-item scale to assess “self-efficacy” was adopted from Chu and Liu [11]. Finally, the five-item scale to assess “perceived clinical barriers” and the three-item construct for “perceived access barriers” were adopted from Coe et al. [26,37]. The ten-item scale to measure “psychological distress” was adopted from Kessler et al. [27,38]. All items were rated from 1 to 7, representing “strongly disagree” to “strongly agree”, respectively. The items of constructs used in the questionnaire was described in Table 2 in detail. 

The demographic information of respondents, including gender, age, monthly income, educational level and marital status, was included in the last section of the questionnaire survey. Moreover, because the target respondents were Vietnamese individuals, all items were first transformed into Vietnamese from the English language. Then, two language experts translated them back into English and compared the two versions to ensure the accuracy of the translation process. 

### 3.2. Sample

Vietnam is considered as an appropriate nation to recruit the data and examine the perceived barriers on intentions to receive COVID-19 vaccines under the mediation and moderation mechanism of self-efficacy, attitude towards COVID-19 vaccines, and psychological distress because the rationales as follows. First, the success of addressing the COVID-19 spread in three first waves by applying 5K (Khẩu trang-mask-wearing, Khử khuẩn-disinfection, Khoảng cách-social distancing, Không tụ tập đông người-No garthering, and Khai báo y tế- health declaration) has been proved in this country [3,12,28,39]. Second, at the time of this conducted research, although the self-development of COVID-19 vaccines (Nanocovax) was conducing in Vietnam, almost all Vietnamese people still did not receive COVID-19 vaccines and waited for the sources of COVID-19 vaccines from international supports. 

The following formula has been used to estimate a minimal size of sample. 

N = 100 + X ∗ Y, [29,40].

With X presenting for even per variable (the recommended X is 50) and Y representing for number of independent variables used in the study (Y = 5). Thus, the minimal sample size was 350. 

The convenient sampling methodology and an online-based survey were used in our study to collect the dataset from 20 July to 20 October 2021 in Vietnam. In this period, Vietnam was undergoing the fourth serious wave of the COVID-19 pandemic [19,41]. Certain restrictions and social distancing measures had been implemented. Therefore, the use of an online-based survey utilizing Google Forms was more appropriate [6,29,30,31]. Participants were clearly informed that their participation was completely voluntary, they could withdraw from the survey at any time, and all information was confidential and only used for academic purposes. In total, 9000 online questionnaires were directly distributed through personal emails, and messages on Facebook, Zalo and Viber, to invite participants to take part in the online-based survey. Only 2913 people participated in our survey. However, 191 responses are eliminated due to containing the missing data. After removing the invalid response, finally, 2722 questionnaires were completely filled, and the response rate reached 30.2%. Table 1 provides the demographic information of respondents. While the largest proportion of respondents were female (56%), most of the respondents were aged 18 to 28 years old and accounted for 60.7%, followed by 29–28 (21.0%), 39–48 (13%), 49–58 (3.8%), and over 59 years old (1.6%). Moreover, 57.7% of respondents earned less than 10 million VND each month; 68.6% of them held a bachelor’s degree, while 66.3% were single. In total, 890 respondents received at least one dose of the COVID-19 vaccine. Finally, the largest proportion of respondents wished to be vaccinated with AstraZeneca (29.1%), followed by Pfizer (19.5%), Moderna (15.3), Sputnik-V (6.6%), Sinopharm (6.8%), Johnson and Johnson (1.9%) and others (Abdala, Hayat-Vax, Janssen, etc.) (14.7%). Interestingly, 6.1% respondents wished to receive Nanocovax, which is a Vietnamese COVID-19 vaccine candidate developed by Nanogen Pharmaceutical Biotechnology JSC.

### 3.3. Analytical Approach

First, univariate normality was evaluated via using skewness and kurtosis values. Second, Cronbach’s alpha, exploratory factor analysis (EFA) and confirmatory factor analysis (CFA) were used to test the reliability and validity of scales. Finally, to estimate both direct and moderation correlations in the conceptual framework, structural equation modeling (SEM) using AMOS 24.0 and the PROCESS macro approach were employed to test the mediation associations [9,42].

## 4. Results

### 4.1. Normality and Scale Assessment

Table 2 illustrates the results of the tests of the normality, reliability and validity of the constructs. First, the normality of scales was evaluated via skewness and kurtosis values, and the results showed that the skewness and kurtosis of all items were within the expected values, as the skewness and kurtosis values of all items were less than 3 and 8, respectively [9,43]. Second, the Cronbach’s alpha values of all scales were higher than 0.63 (the lowest value of Cronbach’s alpha was 0.810), while the corrected item–total correlation of each observed variable was greater than 0.3. Therefore, all scales showed internal consistency and reliability [1]. Whole items were adjusted according to the principal axis factoring method and promax rotation (EFA); however, results showed that the factor loading of ICV2, “I actually get vaccinated for COVID-19”, was lower than 0.5 (λ_ICV2_ = 0.424). After extracting this item, the EFA was re-performed [1]. Results revealed that five factors were loaded with a total extracted variance of 72.454%, whereas the Kaiser–Meyer–Olkin (KMO) value reached 0.939, and the factor loadings of all items were higher than 0.5.

All satisfactory items were adjusted for the CFA, and the results represented a good degree of fitness (see Figure 2). All t-tests of items were significant at the 0.001 degree. Specifically, χ^2^(453) = 4037.916; Chi-square/df = 8.914; *p* < 0.01; GFI = 0.902 > 0.9; AGFI = 0.879 > 0.8; CFI = 0.950 > 0.9; TLI = 0.942 > 0.9; NFI = 0.945 > 0.9 and RMSEA = 0.054 < 0.08 [1,32,44] (Anderson and Gerbing, 1988). In addition, the CFA results also revealed that the standardized regression weights of all observed variables were higher than 0.5 [9,32,33,45].

Additionally, the average variance extracted (AVE) and composite reliability (CR) of all constructs were evaluated (see Table 3). Results showed that the AVE of all constructs was higher than 0.4, while the CR of all constructs was greater than 0.7. Moreover, the square roots of the AVEs of all constructs in the correlation matrix were higher than the inter-constructed association. Thus, the reliability and validity of all scales were satisfactory [9,33].

### 4.2. Structural Equation Modeling

Results of the SEM analysis illustrated that the model reached a high level of fitness. Specifically, χ^2^(287) = 2995.814; Chi-square/df = 10.438; *p* < 0.01; GFI = 0.917 > 0.9; AGFI = 0.891 > 0.8; CFI = 0.945 > 0.9; TLI = 0.932 > 0.9; and RMSEA = 0.059 < 0.8 [1,8,43,45] while the R^2^ (Square Multiple Correlation) of SE, ATC and ICV reached 0.103, 2.39 and 0.607, respectively.

Table 4 reveals the correlations of constructs. First, results indicated that intention to receive COVID-19 vaccines was positively correlated with attitudes towards COVID-19 vaccines (γ = 0.623; *p*-value < 0.001), self-efficacy (γ = 0.060; *p*-value < 0.001) and perceived clinical barriers (γ = 0.142; *p*-value < 0.001), yet it was negatively influenced by perceived access barriers (γ = −0.445; *p*-value < 0.001). Second, self-efficacy was found to have a significant effect on attitudes towards COVID-19 vaccines (γ = 0.158; *p*-value < 0.001). Third, while perceived clinical barriers were positively associated with self-efficacy (γ = 0.677; *p*-value < 0.001) and attitudes towards COVID-19 vaccines (γ = 0.635; *p*-value < 0.001), perceived access barriers were found to have negative impacts on self-efficacy (γ = −0.215; *p*-value < 0.001) and attitudes towards COVID-19 vaccines (γ = −0.445; *p*-value < 0.001).

Regarding the moderation effects of psychological distress, results showed that psychological distress negatively moderated the effect of self-efficacy on the intention to receive COVID-19 vaccines (γ= −0.048; *p*-value < 0.001), and it was found to positively moderate the link between perceived clinical barriers and intention to receive COVID-19 vaccines (γ = 0.079; *p*-value < 0.01). However, psychological distress did not moderate the impacts of attitudes towards COVID-19 vaccines and perceived access barriers on the intention to receive COVID-19 vaccines (*p*-value > 0.05). The results of hypothesis testing were summarized in Table 4.

Figure 3 illustrates the structural equation modeling. In addition, Figure 4 illustrates the interaction plots.

The PROCESS macro with 10,000 bootstrapping samples and a 95% level of confidence was used to test indirect associations (see Table 5). Results showed that attitudes towards COVID-19 vaccines partially mediated the links between self-efficacy (β_indirect SE-ATC-ICV_ = 0.1561; *p*-value < 0.05), perceived clinical barriers (β_indirect PCB-ATC-ICV_ = 0.1958; *p*-value < 0.05), perceived access barriers (β_indirect PAB-ATC-ICV_ = 0.0688; *p*-value < 0.05) and intention to receive COVID-19 vaccines. In addition, self-efficacy partially mediated the effects of perceived clinical barriers (β_indirect PCB-SE-ICV_ = 0.0115; *p*-value < 0.05), perceived access barriers (β_indirect PAB-SE-ICV_ = −0.0043; *p*-value < 0.05) and intention to receive COVID-19 vaccines. Finally, self-efficacy also partially mediated the linkages between perceived clinical barriers (β_indirect PCB-SE-ATC_ = 0.0358; *p*-value < 0.05), perceived access barriers (β_indirect PAB-SE-ATC_ = −0.0124; *p*-value < 0.05) and intention to receive COVID-19 vaccines.

## 5. Discussion

Available information in the literature pertaining to which factors contribute to or limit intentions to receive COVID-19 vaccines is scant. In particular, little is known about the moderation impacts of mental health problems, such as psychological distress, on the links between antecedents and the intention to receive a COVID-19 vaccine. Understanding why individuals express the intention to receive a COVID-19 vaccine is important since it can help health officials and governments in increasing citizens’ awareness of COVID-19 vaccines, promoting vaccination programs and restricting the spread of COVID-19 [13].

Utilizing a sample of 2722 Vietnamese adults, our study adopted structural equation modeling (SEM) to examine the effects of perceived clinical and access barriers on self-efficacy, attitudes towards COVID-19 vaccines and intention to receive COVID-19 vaccines. Concurrently, the moderation effects of psychological distress on these links were also explored. First, our study found that self-efficacy significantly contributed to shaping intentions to receive COVID-19 vaccines. This finding was in contrast with several prior studies [13], but was in line with other studies that explored the links of self-efficacy with other vaccines, such as HPV vaccines [34] or H1N1 influenza vaccination [35,45]. The significant linkage between self-efficacy and intention to receive COVID-19 vaccines may be attributable to the fact that COVID-19 vaccines were available in Vietnam. Second, being consistent with previous research [36], our study also found that attitudes towards COVID-19 vaccines acted as the most important predictor of the intention to receive COVID-19 vaccines. This finding shows that inspiring individuals with favorable/positive attitudes towards COVID-19 vaccines can be considered an effective measure to foster intentions/actions to receive COVID-19 vaccines [13]. Regarding the effects of perceived barriers, while [4] Coe et al. argue that perceived clinical and access barriers to COVID-19 vaccination were not significantly associated with intention to receive COVID-19 vaccines, our study illustrated that both perceived clinical barriers and perceived access barriers were significantly correlated with self-efficacy, attitudes towards COVID-19 and intentions to receive COVID-19 vaccines. However, perceived access barriers were found to have a negative impact on the formation of intentions to receive COVID-19 vaccines, whereas perceived clinical barriers were positively related to intentions to receive COVID-19 vaccines. These outcomes reflect the fact that although people are concerned about the safety or side effects of COVID-19 vaccines, these particular concerns can increase their COVID-19 vaccine behavior, but when they perceive that access to COVID-19 vaccines is difficult or the cost of receiving a COVID-19 vaccine is high, this can reduce their COVID-19 vaccine behavior. For those who, therefore, intend to obtain a COVID-19 vaccine, increasing convenience and providing easier access to COVID-19 vaccines can be identified as an effective strategy to improve COVID-19 vaccination rates [13].

Most importantly, while almost all previous studies neglected the moderation impacts of mental health problems on the path from antecedents to individuals’ receipt of COVID-19 vaccines, our study illustrated that although psychological distress did not increase or reduce the impacts of attitudes towards COVID-19 vaccines and perceived access barriers on the intention to receive COVID-19 vaccines, psychological distress was found to reduce the effect of self-efficacy on intention to receive COVID-19 vaccines. In other words, the translation from self-efficacy into intention to receive COVID-19 vaccines was weakened by psychological distress; although individuals display high levels of confidence regarding COVID-19 vaccines, they hesitate to receive COVID-19 vaccines in case of high psychological distress. Moreover, our study also reveals that the positive impact of perceived clinical barriers on the intention to receive a COVID-19 vaccine can be stronger if the level of psychological distress is high. Thus, this study demonstrated the significant moderating influences of mental health issues on COVID-19 vaccine uptake.

Interestingly, being different from prior studies (e.g., Chu and Liu, 2021; Coe et al., 2021) [11,26], our study not only explored the direct effect of perceived barriers on intention to receive COVID-19 vaccines, but it also examined the mediating role of self-efficacy and attitudes towards COVID-19 vaccines on the links between perceived barriers and intention to receive COVID-19 vaccines. This study found that both self-efficacy and attitudes towards COVID-19 vaccines served as partial mediators in the links between perceived clinical and access barriers and intention to receive COVID-19 vaccines. In other words, perceived clinical and access barriers first significantly affect self-efficacy and attitudes towards COVID-19 vaccines; then, these mediators transfer these impacts to the intention to receive a COVID-19 vaccine. Hence, to increase COVID-19 vaccine receipt and reduce the negative influences of perceived barriers, inspiring favorable attitudes towards COVID-19 vaccines, increasing the availability of COVID-19 vaccines and advancing people’s awareness of COVID-19 vaccines are essential [11,13,36].

## 6. Conclusions

Our study provides several strengths and contributions for both theoretical and practical implications. First, although there have been several scholars investigating COVID-19 vaccine hesitancy/intention, this is one of the earliest studies in Southeast Asia, and it helps to broaden our knowledge on the intention to receive COVID-19 vaccines among adults in Asian countries. Second, the present research is the first to explore the moderation effect of psychological distress on adults’ COVID-19 vaccination. Third, this study found that perceived barriers were significantly involved in the intention to receive COVID-19 vaccines. Additionally, our study provided statistical evidence of the mediating role of attitudes towards COVID-19 vaccines and self-efficacy in the links between perceived barriers and intentions to receive COVID-19 vaccines. Finally, the findings of this study offer a body of useful recommendations for the government, practitioners and policymakers to foster citizens’ COVID-19 vaccine receipt, as well as restrain the spread of the COVID-19 pandemic. Indeed, to increase adults’ COVID-19 vaccine receipt, practitioners and policymakers should focus on effective measures to increase favorable attitudes towards COVID-19 vaccines, make COVID-19 vaccines more available and help people to access COVID-19 vaccines more easily.

However, our study was not without limitations. First, convenient sampling via an online-based survey has been used in our study, although the sample was sizable. Further research should use the random sampling approach to increase the representativeness of the sample. Second, during the period of this study, Vietnam had access to the COVID-19 vaccine, but in limited quantities. As a consequence, this study only focused on exploring adults’ intention to receive COVID-19 vaccines, and a further study should close the intention–behavior link in terms of COVID-19 vaccines and explain why many individuals are hesitant to be vaccinated, as well as investigating how personal and environmental factors influence this link. Third, this study only focused on some antecedents of intentions to receive COVID-19 vaccines, including perceived clinical and access barriers, self-efficacy, and attitudes towards COVID-19 vaccines, further studies can extend the conceptual framework to explore other factors which affect individuals’ intention to receive COVID-19 vaccines. Last, some scales used in our study, which developed by some prior studies, such as Mir et al. [13,32], Chu and Liu [11], and Coe et al. [26]. Although their validity and reliability have been reported through Cronbach’s alpha and CFA, further studies can adopt other scales which can reflect the constructs of perceived clinical and access barriers, self-efficacy, attitudes towards COVID-19 vaccines, psychological distress, and intentions to receive COVID-19 vaccines better. 

## Figures and Tables

**Figure 1 vaccines-11-00289-f001:**
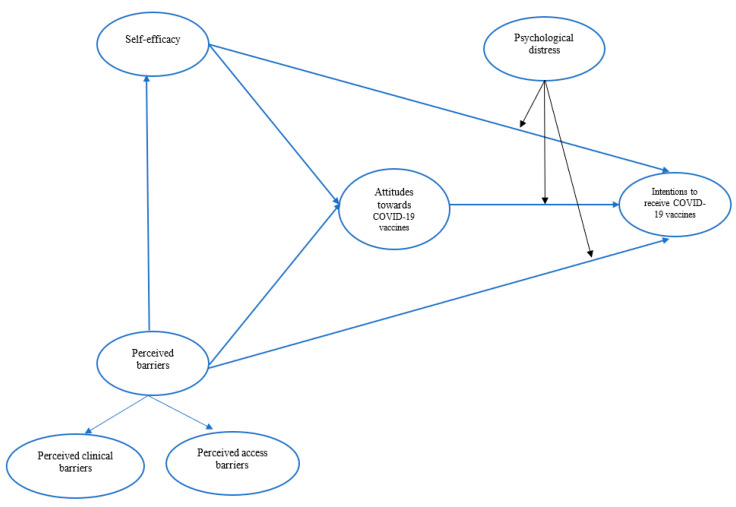
Conceptual framework.

**Figure 2 vaccines-11-00289-f002:**
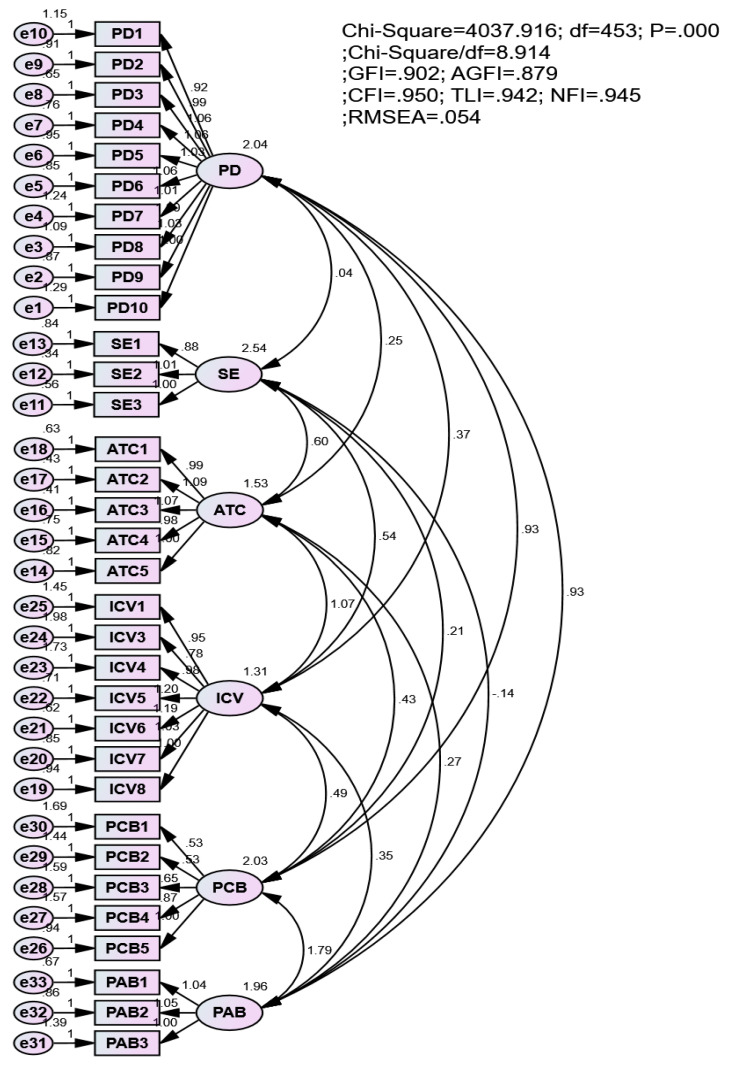
Measurement model.

**Figure 3 vaccines-11-00289-f003:**
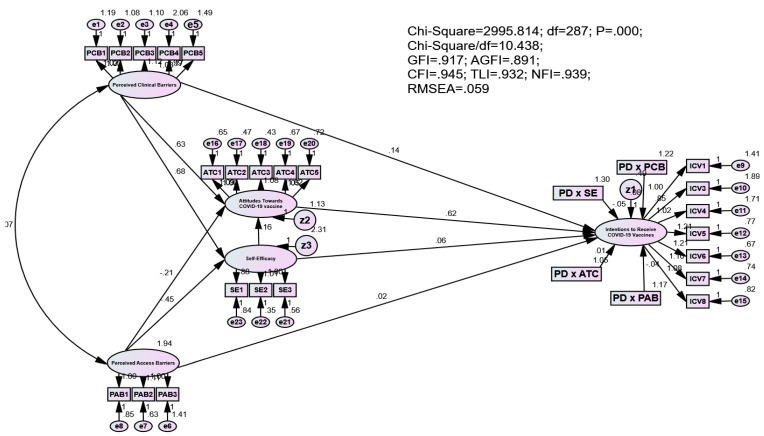
Structural equation modeling.

**Figure 4 vaccines-11-00289-f004:**
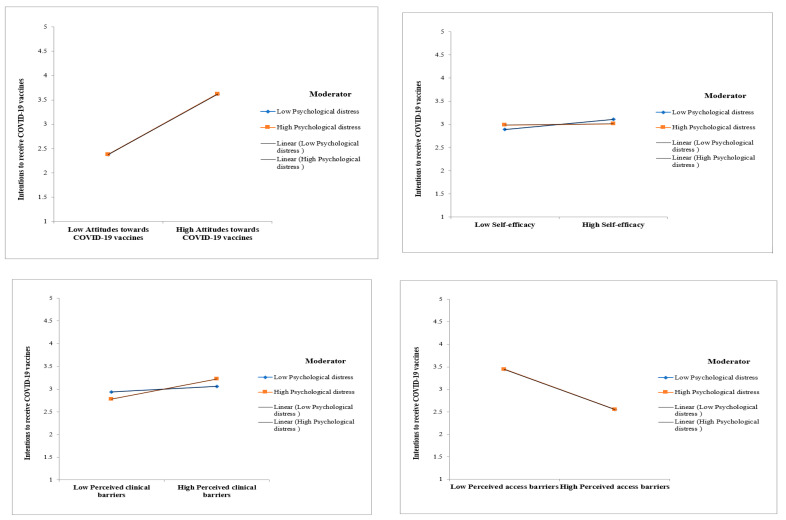
Interaction plots.

**Table 1 vaccines-11-00289-t001:** Demographic characteristics of respondents.

Variables		Frequency	%
Gender	Male	1197	44.0
	Female	1525	56.0
Age	18–28	1651	60.7
	29–38	752	21.0
	39–48	353	13.0
	49–58	103	3.8
	Over 59	43	1.6
Monthly income	Less than 10 million VND	1571	57.7
	From 10 to 20 million VND	668	24.5
	From 20 to 30 million VND	320	11.8
	Over 30 million VND	163	6.0
Educational level	High school	597	21.9
	Bachelor’s degree	1867	68.6
	Master’s/PhD degree	258	9.5
Marital status	Single	1804	66.3
	Married	918	33.7
Did you receive a COVID-19 vaccine?	Yes	890	32.7
	Not yet	1832	67.3
Please choose a type of COVID-19 vaccines in which you would like to be vaccinated?	AstraZeneca	793	29.1
	Pfizer	532	19.5
	Moderna	416	15.3
	Sputnik-V	179	6.6
	Sinopharm	185	6.8
	Johnson and Johnson	52	1.9
	Nanocovax	166	6.1
	Others (Abdala, Hayat-Vax, Janssen, …)	399	14.7

Note: N = 2722.

**Table 2 vaccines-11-00289-t002:** Cronbach’s alpha, pattern matrix and descriptive characteristics of variables.

Code	Variables	Mean	SD	Skewness	Kurtosis	Pattern Matrix (EFA)						Factor Loading (CFA)
						F1	F2	F3	F4	F5	F6	
PD	Psychological distress [13] (Cronbach’s alpha α = 0.968)	3.5134	1.49749	0.285	−0.859							
PD1	I often feel tired out for no good reason	3.6146	1.70163	0.242	−0.973	0.785						0.776
PD2	I often feel nervous	3.7381	1.70545	0.072	−1.105	0.848						0.828
PD3	I often feel so nervous that nothing could calm me down	3.3472	1.71777	0.401	−0.834	0.837						0.883
PD4	I often feel hopeless	3.2164	1.74879	0.521	−0.759	0.792						0.866
PD5	I often feel restless or fidgety	3.5393	1.76185	0.238	−1.061	0.867						0.833
PD6	I often feel so restless that I could not sit alone	3.4361	1.77381	0.333	−1.008	0.865						0.854
PD7	I often feel depressed	3.7961	1.82391	0.104	−1.145	0.837						0.791
PD8	I often feel that everything is not effort	3.5496	1.77282	0.291	−1.000	0.856						0.807
PD9	I often feel so sad that nothing could cheer me up	3.3982	1.74487	0.403	−0.867	0.824						0.845
PD10	I often feel worthless	3.4989	1.82337	0.313	−1.016	0.823						0.783
ICV	Intention to receive COVID-19 vaccines [3,17] (Chu and Liu, 2021; Mir et al., 2021) (Cronbach’s alpha α = 0.896)	4.9007	1.25748	−0.630	0.042							
ICV1	I try to get COVID-19 vaccines	4.8916	1.62747	−0.620	−0.438		0.625					0.672
ICV3	I get vaccinated if a physician offered me COVID-19 vaccines	4.5882	1.66920	−0.356	−0.769		0.583					0.538
ICV4	I intend to take up COVID-19 vaccines soon	4.5206	1.72954	−0.436	−0.762		0.705					0.650
ICV5	I would recommend my family members to take up COVID-19 vaccines	4.9375	1.60935	−0.631	−0.361		0.884					0.853
ICV6	I intend to take up COVID-19 vaccines if it is recommended by a doctor	5.0004	1.57117	−0.678	−0.198		0.899					0.865
ICV7	I intend to take up COVID-19 vaccines as it boots my immune system	5.1209	1.49718	−0.754	0.018		0.779					0.788
ICV8	I have the firm intention to receive a COVID vaccine	5.2454	1.50071	−0.814	0.145		0.701					0.763
ATC	Attitude towards receiving a COVID-19 vaccine [17] (Mir et al., 2021) (Cronbach’s alpha α = 0.933)	5.1947	1.32337	−0.909	0.419							
ATC1	In my opinion, COVID-19 vaccine is an effective response to the corona pandemic	5.2300	1.45974	−0.854	0.321			0.709				0.840
ATC2	I have surely decided to take up COVID-19 vaccines	5.2777	1.49826	−0.872	0.190			0.796				0.899
ATC3	I would never refuse a dose of COVID-19 vaccines	5.2506	1.47341	−0.796	0.125			0.811				0.900
ATC4	I think COVID-19 vaccines are a necessity for all people	5.0764	1.48941	−0.655	−0.118			0.794				0.813
ATC5	Receiving a COVID-19 vaccine implies more advantages than disadvantages to me	5.1389	1.53119	−0.743	−0.076			0.775				0.808
SE	Self-efficacy [3] (Chu and Liu, 2020) (Cronbach’s alpha α = 0.922)	4.2531	1.59601	−0.294	−0.724							
SE1	I will be able to get the vaccines to prevent contracting COVID-19	4.4195	1.68122	−0.367	−0.692				0.797			0.838
SE2	I will be easy for me to get the vaccines to protect myself from COVID-19	4.2120	1.70615	−0.215	−0.818				0.944			0.939
SE3	Getting vaccinated to prevent COVID-19 is convenient	4.1278	1.76135	−0.183	−0.871				0.916			0.905
PCB	Perceived Clinical Barriers [4] (Coe et al., 2021) (Cronbach’s alpha α = 0.810)	4.4019	1.21497	−0.301	−0.227							
PCB1	I will have side effects from the COVID-19 vaccine	4.8424	1.50454	−0.581	−0.294					0.768		0.501
PCB2	The COVID-19 vaccine will be safe	4.8880	1.48332	−0.653	−0.163					0.833		0.529
PCB3	I will get sick from the COVID-19 vaccine	4.7417	1.56184	−0.478	−0.488					0.656		0.590
PCB4	I will die from the COVID-19 vaccine	3.5709	1.76628	0.227	−0.963					0.708		0.705
PCB5	The COVID-19 vaccine will be painful	3.9666	1.72428	−0.051	−1.040					0.751		0.827
PAB	Perceived access barriers [4] (Coe et al., 2021) (Cronbach’s alpha α = 0.865)	3.9040	1.54946	0.035	−0.945							
PAB1	It will be hard for me to get the COVID-19 vaccine	4.1705	1.67091	−0.106	−0.935						0.737	0.872
PAB2	There will not be enough of the COVID-19 vaccine for me	3.9445	1.73307	0.037	−1.033						0.847	0.846
PAB3	The COVID-19 vaccine will cost me a lot of my own money	3.5970	1.83066	0.181	−1.109						0.830	0.764

Note: N = 2722, EFA: Exploratory factor Analysis, CFA: Confirmatory Factor Analysis.

**Table 3 vaccines-11-00289-t003:** Correlation matrix, the reliability and discriminant validity of constructs.

	CR	AVE	PCB	PD	SE	ATC	ICV	PAB
PCB	0.772	0.412	0.642					
PD	0.956	0.684	0.364 **	0.827				
SE	0.923	0.801	0.126 **	0.008	0.895			
ATC	0.930	0.728	0.304 **	0.165 **	0.296 **	0.853		
ICV	0.893	0.549	0.361 **	0.249 **	0.293 **	0.698 **	0.741	
PAB	0.868	0.687	0.666 **	0.427 **	−0.048 *	0.131 **	0.229 **	0.829

Notes: N = 2722, **: Significance at 0.01 level (two-tailed); *: Significance at 0.05 level (two-tailed); AVE: Average Variance Extracted; CR: Composite Reliability. The diagonal elements (in bold): The square root of the AVE of each construct. ICV = Intention to receive COVID-19 vaccine, ATC = Attitudes towards COVID-19 vaccines, SE = Self-efficacy, PCB = Perceived clinical barriers; PAB = Perceived access barriers; PD = Psychological distress.

**Table 4 vaccines-11-00289-t004:** Correlations between constructs.

	Correlations			Estimate	S.E.	C.R.	*p*-Value	Results
H1	ATC	→	ICV	0.623	0.022	28.676	***	Supported
H2a	SE	→	ICV	0.060	0.011	5.443	***	Supported
H2b	SE	→	ATC	0.158	0.016	10.036	***	Supported
H3a	PCB	→	ICV	0.142	0.037	3.816	***	Supported
H3b	PCB	→	ATC	0.635	0.046	13.846	***	Supported
H3c	PCB	→	SE	0.677	0.059	11.532	***	Supported
H4a	PAB	→	ICV	−0.445	0.042	−10.577	***	Supported
H4b	PAB	→	ATC	−0.215	0.031	−6.823	***	Supported
H4c	PAB	→	SE	−0.445	0.042	−10.577	***	Supported
H5a	PD × ATC	→	ICV	0.007	0.017	0.426	0.670	Not supported
H5b	PD × SE	→	ICV	−0.048	0.014	−3.319	***	Supported
H6a	PD × PCB	→	ICV	0.079	0.025	3.213	0.001	Supported
H6b	PD × PAB	→	ICV	−0.038	0.024	−1.588	0.112	Not supported

Notes: N = 2722, *** *p* < 0.001, ICV = Intention to receive COVID-19 vaccine, ATC = Attitudes towards COVID-19 vaccines, SE = Self-efficacy, PCB = Perceived clinical barriers; PAB = Perceived access barriers; PD = Psychological distress.

**Table 5 vaccines-11-00289-t005:** The mediation coefficients.

*Mediation* Standardized *Regression Coefficients*					Indirect Effects	SE	95% Confidence Interval	
							LLCI	ULCI
SE	→	ATC	→	ICV	0.1561 *	0.0124	0.1316	0.1800
PCB	→	ATC	→	ICV	0.1958 *	0.0163	0.01637	0.2281
PAB	→	ATC	→	ICV	0.0688 *	0.0109	0.0479	0.0903
PCB	→	SE	→	ICV	0.0115 *	0.0029	0.0064	0.0177
PAB	→	SE	→	ICV	−0.0043 *	0.0020	−0.0084	−0.0004
PCB	→	SE	→	ATC	0.0358 *	0.0070	0.0228	0.0501
PAB	→	SE	→	ATC	−0.0124 *	0.0057	−0.0235	−0.0015

Notes: N = 2722; LLCI: Lower level of confidence interval. ULCI: Upper level of confidence interval. SE: Standard errors. * *p* < 0.05. ICV = Intention to receive COVID-19 vaccines, ATC = Attitudes towards COVID-19 vaccines, SE = Self-efficacy, PCB = Perceived clinical barriers; PAB = Perceived access barriers.

## Data Availability

The datasets collected and analyszed during the research are not publicy available because of the some sensitive information, such as demographic profiles and personal perceptions. Nevethless the dataset and testing results reported from SPSS 28.0, and AMOS 25.0 codes are available from the research team if a reasonable requirement.

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
