# Peer review of "Perceived Barriers and Intentions to Receive COVID-19 Vaccines: Psychological Distress as a Moderator"

_vaccines, 2023, doi:10.3390/vaccines11020289_

Round 1

Reviewer 1 Report

intresting papaers that meets the scope of the journal. Clear structure, sound methodology, proper argumentation supported by recent bibliography.

Appropriate writting tone, fine language grammar ans syntax.

Referencing system acceptable

Author Response

Dear sir/madam,

Thanks again for your support. I send you the response to the comments in the attached files.

Thank you very much. You suggested the very useful comments. I think your advice make our paper more better. I hope that you feel well with my revised article.

I am looking forward to hearing from you soon.

Best regards,

Prof. Dr. Vu Ngoc Xuan

National Economics University

Reviewer 2 Report

The article presented for review is related to a very important aspect of the COVID-19 pandemic, namely the psychological mechanisms influencing the decision to accept the vaccine. The experience of the last two years shows that psychological factors play an important role in this matter.

The article has a correct structure, the purpose of the work is included in the Introduction. The description of the tests is correct and the method of conducting the research is clear. The group is numerous and well characterized socio-demographically. The techniques used are correct, and the use of structural equation modeling deserves special attention. The model adopted in the study is correct and clear and adequately analyzed statistically.

I have three minor comments: I think it would be interesting for the reader to provide in the Introduction data on the number of cases in Vietnam and data on the number of vaccinated. Not all data on the diagrams are legible, maybe some can be removed without damaging the substantive value of the models? In the statistical part concerning correlation analyses, I am concerned about taking into account - in the analysis and discussion of the results - extremely low values of the correlation coefficients, for example -0.048. This exemplary coefficient is of course statistically significant (which may be due to the fact that the group is large in size), but inference should be very careful.

Author Response

(The authors gave the same response as above.)

Reviewer 3 Report

Comments and Questions for vaccines-2155843

1. Introduction

However, very limited studies have been conducted 54 to explore the issues of vaccination intention and hesitancy in Southeast Asian countries, such as Vietnam”. This statement shows that the authors have not been able to review sufficient studies in Vietnam. For example:

https://doi.org/10.3390/vaccines10020222

https://doi.org/10.1007/s11482-022-10104-5

https://doi.org/10.3390/vaccines10111775

2. Conceptual framework

References are needed for the conceptual framework.

The authors are encouraged to describe factors illustrated in the conceptual framework in detail, for example, construction, and measures/units. It is not easy to imagine how high/low the level of distress is. Similarly, How to define a respondent who faced a clinical/access barrier is not clear.

The authors are expected to develop hypotheses on the impact of psychological distress on COVID-19 vaccine acceptance.

Regression models should be clearly presented and described.

Variable selection with expectations should be clearly addressed.

3. Questionnaires

The authors are suggested to provide the questionnaires in the Appendix/ices.

4. Informed consent and ethical approval

How the respondents’ informed consent was obtained? If the informed consent was not applicable, how reliable the responses were?

Since this study involves human beings' participation, please tell us: how was the ethical approval was obtained and by whom? Please provide the following information: ethical approval Board, ID/protocol. If ethical approval was not obtained, how reliable the /surveys/responses were? Please explain why?

5. Sample

How well sample represented the study location/country (Vietnam)?

6. Methodologies and results

The scales addressed from lines 74 to 82 should be described in detail. For example, what are the first 3 items?

What criterion/a was/were used to group Janssen with other vaccines and why?

COVID-19 vaccine Nanocovax was not approved at the time of the study (20 Jul to 20 Oct 2021) (https://baochinhphu.vn/ket-qua-giua-ky-thu-nghiem-lam-sang-giai-doan-3-vaccine-nanocovax-102305810.htm), why it was used in the study?

A number of influential factors (For example, the levels of respondents’ exposure to COVID-19/SARS-CoV-2 virus, and respondents’ perception of COVID-19 pandemic/SARS-CoV-2 virus) have not been taken into account. How reliable the study results are if these factors are excluded?

I will be easy for me to get the vaccines to protect myself from COVID-19
”. A such sentence is not easy to understand. How could the respondents understand to participate in the survey?

these particular concerns can increase their COVID-19 vaccine behavior”. It is common to judge behaviour as good/bad or appropriate/inappropriate, but it is hard to imagine that behaviour can be increased. The authors are encouraged to explain this.

These outcomes reflect the fact that although people are concerned about the safety or side effects of COVID-19 vaccines, these particular concerns can increase their COVID-19 vaccine behavior, but when they perceive that access to COVID-19 vaccines is difficult or the cost of receiving a COVID-19 vaccine is high, this can reduce their COVID-19 vaccine behavior.” Since the access to COVID-19 vaccines and the vaccine cost were grouped, particularly, these variables were not separately regressed, how to examine the impact of each variable on the willingness to vaccinate? Especially, COVID-19 vaccines have been freely given in Vietnam. Similar comments are given to similar groups/variables.

7. English

English can be further improved. For example, “The significant linkage between self-efficacy and intention to receive COVID-19 vaccines may be attributable the fact that COVID-19 vaccines were available in Vietnam”. A preposition “to” is expected after “attributable” and before “the”.

Author Response

(The authors gave the same response as above.)

Round 2

Reviewer 3 Report

Thank you for your responses. The authors are responsible for ensuring that ethical approval has been approved and obtained. Please rephrase the information that has been added as currently, it overlaps with some of the prior studies. In addition, please refer to the original studies to cite. Also, those that can not be fixed/improved (refer to the previous comments) should be acknowledged as limitations and clearly addressed in the "Study/research limitation".

Best regards,

Author Response

(The authors gave the same response as above.)
